# Overuse of computed tomography for mild head injury: A systematic review and meta-analysis

**Maryam Saran[1], Morteza Arab-Zozani[2], Meysam Behzadifar[1], Mehrdad Gholami[3], Samad Azari[4], Nicola Luigi Bragazzi[5], Masoud Behzadifar[1] ***

**1** Social Determinants of Health Research Center, Lorestan University of Medical Sciences, Khorramabad, Iran, **2** Social Determinants of Health Research Center, Birjand University of Medical Sciences, Birjand, Iran, **3** Department of Medical Physics, School of Medicine, Lorestan University of Medical Sciences, Khorramabad, Iran, **4** Hospital Management Research Center, Health Management Research Institute, Iran University of Medical Sciences, Tehran, Iran, **5** Human Nutrition Unit Department of Food and Drugs, University of Parma Medical School, Parma, Italy

* masoudbehzadifar@gmail.com; behzadifar@lums.ac.ir

## Abstract

### Background

Computed tomography (CT) scan is a common imaging technique used to evaluate the severity of a head injury. The overuse of diagnostic interventions in the health system is a growing concern worldwide.

**Objectives:** The aim of this systematic review is to investigate the rate of CT scan overuse in cases of mild head injury.

### Methods

**Eligibility criteria:** We encompassed observational studies—either designed as cohort, case-control, or cross-sectional investigations—that reported on CT scan overuse rates for mild head injuries. Studies had to be published in peer-reviewed, English-language sources and provide full content access

**Information sources:** Web of Sciences, Scopus, Medline *via* PubMed, the Cochrane Library and Embase were searched from inception until April 1, 2023. Studies were included if reporting the overuse of CT scans for mild head injuries using validated criteria.

**Risk of bias:** We used the Risk Of Bias In Non-randomised Studies - of Interventions (ROBINS-I) tool to evaluate the risk bias assessment of included studies. Two independent reviewers evaluated the eligibility of studies, extracted data, and assessed study quality by using the Newcastle-Ottawa Scale.

**Synthesis of results:** Overuse estimates were calculated using a random-effects model. Subgroup analyses were performed to investigate any sources of heterogeneity. Point rate of overuse of CT scans for mild head injuries was the main outcome measured as percentage point estimates with corresponding 95% CIs.

**Data Availability Statement:** All relevant data are within the manuscript and its Supporting Information files.

**Funding:** The author(s) received no specific funding for this work.

**Competing interests:** The authors have declared that no competing interests exist.

**Abbreviations:** CT, Computed tomography; MRI, Magnetic resonance imaging; LMICs, Low- and middle-income countries; PRISMA, Preferred Reporting Items for Systematic Review and Meta-Analyses statement; NOS, Newcastle-Ottawa Scale; CI, Confidence intervals; GCS, Glasgow Coma Scale; **CCH**R, Canadian computed tomography Head Rule; PECARN, Pediatric Emergency Care Applied Research Network; NICE, National Institute for Health and Clinical Excellence; NOC, New Orleans Criteria; RD, Risk Difference.

# Results

**Included studies:** Of the 913 potentially relevant studies identified, eight studies were selected for the final analysis.

**Synthesis of results:** The pooled rate of CT scan overuse in patients with mild head injury was found to be 27% [95% CI: 16–43; $I^2$ = 99%]. The rate of CT scan overuse in mild head injury cases varied depending on the criteria used. The rate of CT scan overuse was 37% [95% CI: 32–42; $I^2$ = 0%] with the Glasgow Coma Scale (GCS), 30% [95% CI: 16–49; $I^2$ = 99%] with the Canadian computed tomography head rule, and 10% [95% CI: 8–14; $I^2$ = 0%] with the Pediatric Emergency Care Applied Research Network criterion (PERCAN). Based on subgroup analyses, the rate of CT scan overuse in mild head injury cases was observed to be 30% with the Canadian computed tomography head rule criterion, 43% with the National Institute for Health and Clinical Excellence criterion, and 18% with the New Orleans criterion.

# Conclusion

**Limitations of evidence:** The restricted number of included studies may impact generalizability. High heterogeneity was observed, leading to subgroup analyses based on age, assessment criteria, and study region. Absent data on overuse causes hinders drawing conclusions on contributing factors. Furthermore, this study solely addressed overuse rates, not associated harm or benefits.

**Interpretation:** The overuse of CT scans in mild head injury patients is concerning, as it can result in unnecessary radiation exposure and higher healthcare costs. Clinicians and policymakers should prioritize the implementation of guidelines to reduce unnecessary radiation exposure, healthcare costs, and potential harm to patients.

# Trial registration

The study protocol of this review was registered in PROSPERO under the identification code CRD42023416080. https://www.crd.york.ac.uk/prospero/display_record.php?ID=CRD42023416080.

# Introduction

Mild head injury is a common type of injury that affects the brain. It occurs when a blow or jolt to the head causes the brain to move back and forth rapidly, resulting in a disruption of normal brain function [1]. While mild head injury is typically not life-threatening, it can cause a wide range of physical, cognitive, and emotional symptoms that can last for days, weeks, or even months [2].

Computed tomography (CT) scan is a common imaging technique used to evaluate head injuries, especially in cases of mild head injuries [3,4]. It helps identify structural damages like bleeding or swelling, which could be life-threatening if not promptly treated [5]. CT scans also aid in determining the severity of the injury and guiding the appropriate management plan [6]. However, it is essential to exercise clinical judgment and consider individual circumstances, as CT scans may not always be necessary for mild head injuries without loss of consciousness or

concerning symptoms [7,8]. Yet, for severe symptoms or injuries from high-impact activities, a CT scan is typically recommended to rule out underlying structural damage [9,10].

The overuse of diagnostic interventions in the health system is a growing concern worldwide [11]. The widespread use of diagnostic interventions such as CT scans, magnetic resonance imaging (MRI), and other imaging techniques, can result in unnecessary healthcare spending and increase the risk of harm to patients [12]. Overuse can also lead to a false sense of security and delay the proper diagnosis and treatment of patients who actually require these interventions [13]. Overuse of diagnostic interventions is not only a problem in developed countries, but it is also an issue in low- and middle-income countries where resources are limited [14]. In these settings, limited access to diagnostic interventions can result in healthcare providers relying on clinical judgment and physical examination to make a diagnosis [15,16]. However, the lack of access to diagnostic interventions can also result in healthcare providers relying on guesswork or assumptions, which can lead to misdiagnosis and delayed treatment. Overuse of CT in cases of mild head injury is a significant concern as it can lead to unnecessary exposure to radiation and an increase in healthcare costs [11]. The aim of this systematic review is to investigate the rate of CT scan overuse in cases of mild head injury. The findings of this study will facilitate enhanced adherence to guidelines and may assist in the development and implementation of revised treatment protocols.

## Objectives

The overuse of CT scans in cases of mild head injuries in healthcare systems can lead to unnecessary expenses and potential harm to patients. The study aims to raise awareness about the evidence-based and judicious use of imaging techniques, emphasizing the importance of understanding the prevalence of CT scan overuse in mild head injuries in order to improve clinical decision-making. By implementing evidence-based guidelines, patient safety can be enhanced, avoiding unnecessary radiation exposure, and reducing false positives or overdiagnosis. The systematic review will provide insights into healthcare providers' adherence to existing guidelines, helping develop revised treatment protocols aligned with evidence-based practices. The investigation can address potential diagnostic delays in cases where CT scans are genuinely necessary, ensuring timely treatment and better patient outcomes. Despite the importance of appropriate imaging in mild head injuries, comprehensive studies on global CT scan overuse prevalence are lacking.

This review and meta-analysis aims to fill this knowledge gap by synthesizing evidence from various countries, making a valuable contribution to the field's literature.

## Methods

This study is reported according to the Preferred Reporting Items for Systematic Review and Meta-Analyses statement (PRISMA) (**S1 and S2 Tables**) [17].

### Eligibility criteria

**Inclusion Criteria**

The inclusion criteria were as follows:

1. Study Design: We considered observational studies, including cohort studies, case-control studies, and cross-sectional studies, which reported on the prevalence or rates of overuse of CT scans for mild head injuries.

2. Publication Source: Studies had to be published in peer-reviewed, scholarly journals.

3. Language: Only studies published in the English language were included to facilitate data extraction and analysis.

4. Availability: Studies had to have their entire content available, either as open-access publications or through institutional access, to ensure unrestricted access to relevant data.

**Exclusion Criteria:**
The exclusion criteria were as follows:

1. Incomplete Data: Studies lacking complete data to estimate the overall rate of CT overuse were excluded from our analysis. We aimed to include studies with sufficient information for meaningful knowledge synthesis and meta-analysis.

2. Review Articles: Studies that were published as review articles were excluded from our analysis to avoid duplicated data already presented elsewhere and to focus on original research.

3. Limited Availability: Studies without full-text availability were excluded to ensure proper evaluation of their methodology and results.

4. Conference Abstracts: We did not consider conference abstracts, as they often lack sufficient detail for comprehensive analysis and are more prone to selection bias. Even if, on the one hand, we recognize the importance of considering valuable information from gray literature sources, to ensure a comprehensive review of the literature, on the other hand, this may compromise the quality of our review. As such, we decided not to conduct a supplementary search specifically targeting gray literature. This search would have allowed us to have access to potentially relevant studies that might not have been captured in the peer-reviewed journal databases. The process of searching for gray literature, indeed, involves utilizing various platforms and databases, such as institutional repositories, conference websites, and relevant government databases. However, gray literature may lack quality and relevance to our research topic.

5. Unpublished Manuscripts: Unpublished manuscripts were excluded to maintain the integrity and verifiability of the included studies.

6. Interventional Studies: Studies with an intervention-based design were excluded, as our focus was on observational studies that reflected real-world clinical practices and trends.

## Information sources

**Search strategy.** We conducted a systematic search for relevant studies through electronic databases including Web of Sciences, Scopus, Medline *via* PubMed, the Cochrane Library, and Embase. The search was limited to articles published from inception until April 1, 2023, and involved combining specific terms such as "prevalence," "overuse, "rate", and "mild head injury." Two teams (MS, MG, and MAZ, MeB), each consisting of two researchers, conducted the searches independently, and any discrepancies were resolved through discussion. In addition, we examined the reference lists of identified articles and used Google Scholar. There were no geographic restrictions applied to the search. (**The search strategies used for each database can be found in S3 Table**).

## Data collection process and data items

Following the selection of relevant articles, two authors (NLB, SA) extracted information, including the first author's name, publication year, country, criteria for diagnosing acute head

injury, number of participants, age range or average, gender (number of male/female subjects), criteria for CT scan performance, and prevalence of overuse. Conflicting results were resolved by a senior (MS) researcher, and if disagreements persisted, a third reviewer (MaB) was introduced to reach a consensus. Data collection followed a form approved and designed by the authors' group (MAZ, MeB, and MG). In instances where data was incomplete or the full text was unavailable, we contacted the corresponding author.

### Study risk of bias assessment

We used the "Risk Of Bias In Non-randomised Studies - of Interventions" (ROBINS-I) tool to evaluate the risk bias assessment of included studies in our assessment [18]. The following domains were examined with this tool: (1) bias resulting from confounding factors, (2) bias in participant selection for the study, (3) bias in the categorization of interventions, (4) bias due to deviations from the intended interventions, (5) bias resulting from missing data, (6) bias in the measurement of outcomes, and (7) bias in the selection of reported results. Two authors (MS, NLB) performed this assessment independently, and a third author (MaB) resolved potential disagreements.

### Quality assessment of included studies

Quality assessment was assessed for each study using the Newcastle-Ottawa Scale (NOS), with any discrepancies among authors resolved through consensus. The Newcastle–Ottawa scale is a tool used for assessing the quality of non-randomized studies. Studies were categorized based on their NOS score, with scores of 1–3 indicating high quality, scores of 4–6 indicating moderate quality, and scores of 7–9 indicating low quality [19]. Two authors (SA, MG) performed this activity independently, and a third author (MS) resolved disagreements between them.

### Synthesis methods

We used the R software Version 4.2.3 utilizing the *meta* package to perform the meta-analysis. The random-effects model was used to calculate the pooled rate, with the DerSimonian-Laird approach applied to compute 95% Confidence Intervals (CI). Heterogeneity among studies was assessed using the $I^2$ statistic. We used Baujat plot to explore heterogeneity. To assess the impact of each study, sensitivity analysis was conducted by omitting each study sequentially. Additionally, to ensure the credibility of the sensitivity analysis outcomes, a cumulative meta-analysis was performed to examine the impact of study order. We used a qualitative assessment to assess publication bias. We visually inspected the funnel plot and employed Egger's test to quantitatively confirm the presence of small-study effects. As publication bias was identified, we utilized Duval and Tweedie's non-parametric/trim and fill method to adjust the combined estimate. Two-sided *P* values were statistically significant at less than 0.05.

## Results

### Study selection

We identified 913 records initially, and after duplicate removal, 532 records were sought for screening. We screened the titles and abstracts of these articles and excluded 479 records. Then, we evaluated the full text of the remaining 53 records for eligibility, and 45 were excluded. Finally, we included 8 studies in our analysis, as shown in **Fig 1** [20–27].

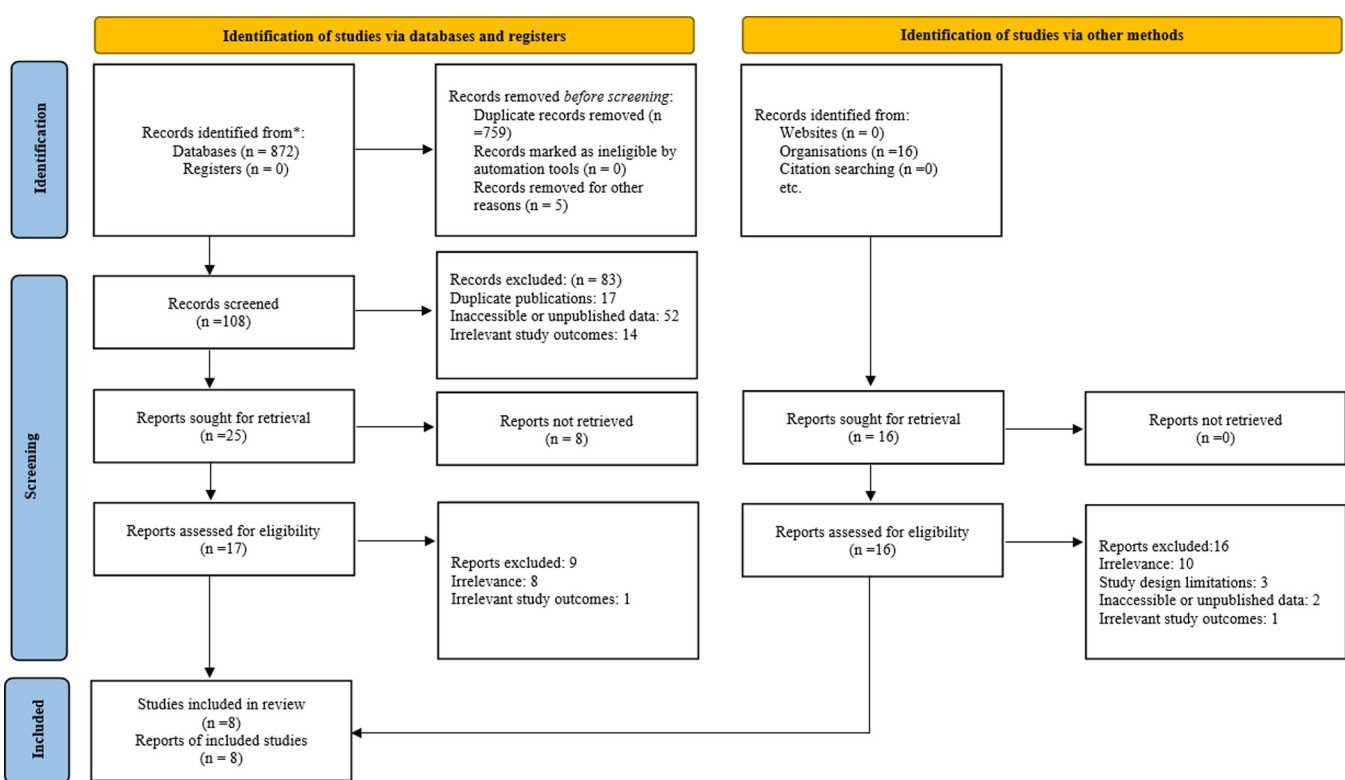

**Fig 1. Preferred Reporting Items for Systematic Reviews and Meta-Analyses (PRISMA) flow diagram of the search, screen and selection of studies focusing on the overuse of computed tomography in mild head injury.** Of the 913 potentially relevant studies identified, eight studies were selected for the final analysis.

## Study characteristics

Summary characteristics of the included studies, involving a total of 3605 participants, are shown in Table 1.

## Quality assessment

One study [24] received a score of 6, two studies [21,22] scored 7, four studies [23,25–27] scored 8, and one study [20] scored 9. Of these, 12.5% of the studies were found to have a moderate risk of bias, and 87.5% were to have a low risk of bias (**S4 Table**).

**Table 1. The characteristics of the selected studies.**

| First author (Reference) | Year | Country | Criteria for CT scan performance | Mean age | Male (%) | Female (%) | Sample size |
|---|---|---|---|---|---|---|---|
| Melnick [20] | 2012 | USA | CCHR | 48 | 215 (62.1) | 131(37.9) | 346 |
| Zargar Balaye Jame [21] | 2014 | Iran | Glasgow Coma Scale | 36.9 ± 19.6 | 228 (57) | 172 (43) | 400 |
| Klang [22] | 2016 | Israel | CCHR | 69.4±21.8 | 428 (44.82) | 527 (55.18) | 955 |
| Cellina [23] | 2018 | Italy | CCHR | 32 ± 3 | NA | NA | 493 |
| Tan [24] | 2018 | Singapore | CCHR | 48 | 218 (62.4) | 131 (37.6) | 349 |
| Gariepy [25] | 2019 | Canada | PECARN | NA | 240 (59.11) | 166 (40.89) | 406 |
| Shobeirian [26] | 2020 | Iran | CCHR | 38.38 ± 19.73 | NA | NA | 170 |
| Al Omran [27] | 2023 | Bahrain | CCHR | 44.86 ± 21.609 | 331 (68.1) | 155 (31.9) | 486 |

(CCHR: Canadian Computed Tomography Head Rule, PECARN: The Pediatric Emergency Care Applied Research Network, NA: Not applicable).

## Risk of bias

Fig 2 shows the summary of bias risk assessment for the studies included in the analysis. For bias related to confounding and bias in participant selection, all studies demonstrated low risk. Regarding bias in the classification of interventions, one study exhibited low risk, six studies had uncertain risk, and one study showed high risk. As for bias due to deviations from intended interventions, four studies had low risk, three studies had unclear risk, and one study had high risk. For bias stemming from missing data and bias in outcome measurement, six studies showed low risk, and two studies had unclear risk. Finally, concerning bias in the selection of reported results, seven studies displayed low risk, while one study had unclear risk.

## Synthesis of results

The overall rate of CT overuse in mild head injury was estimated to be 27% [95% CI: 16–43; $I^2$ = 99%]. The rate of CT overuse in mild head injury, as determined by the physician's decision criteria in the emergency unit used for patient evaluation, is displayed in Fig 3. The results indicate that the rate of CT overuse in mild head injury was 37% [95% CI: 32:42; $I^2$ = 0%] using the Glasgow Coma Scale (GCS), 30% [95% CI: 16:49; $I^2$ = 99%] using the Canadian Computed Tomography Head Rule (CCHR) criterion, and 10% [95% CI: 8:14; $I^2$ = 0%] using the Pediatric Emergency Care Applied Research Network (PECARN) criterion.

   Due to the substantial amount of heterogeneity observed among the studies, Baujat plot was drawn (Fig 4). This visual representation demonstrates that most studies had similar rates of overuse, but the Cellina study [23] had notably a higher rate compared to the others. When we excluded this study from our analysis, the overuse rate was estimated to be 22% [95% CI: 15–32; $I^2$ = 98%].

## The sensitivity and cumulative analysis

To ensure the robustness of the sensitivity analysis, the consistency of the impact of each study was assessed before and after its removal from the analysis (Fig 5). Furthermore, the effect of study order was examined through cumulative meta-analysis. The result of these two analyses showed that the rate of CT overuse in mild head injury was 27% [95% CI: 16–43; $I^2$ = 99%], (Fig 6).

## Subgroup analysis

The rate of CT overuse in mild head injury patients was 36% [95% CI: 3–92; $I^2$ = 100%] in Europe, 27% [95% CI: 20–36; $I^2$ = 91%] in Asia, and 20% [95% CI: 5–53; $I^2$ = 98%] in America. Additionally, the rate of CT overuse in mild head injury patients was 36% [95% CI: 14–66; $I^2$ = 99%] in individuals aged 40 years and below, and 20% [95% CI: 13–30; $I^2$ = 92%] in those above 40 years. Compliance with clinical decision rules to use CT, in cases of mild head injury the rate of CT overuse was observed to be 30% [95% CI: 16: 49] in the CCHR, 43% [95% CI: 13: 80] in the National Institute for Health and Clinical Excellence (NICE), and 18% [95% CI: 5:45] in the New Orleans Criteria (NOC). (S1–S6 Figs). The rate of CT Overuse of scan in patients with mild head injury was 35% [95% CI: 30:40] in America, 35% [95% CI: 30:40] in Iran, 23% [95% CI: 19:27] in Bahrain, 20% [95% CI: 16:25] in Singapore, 11% [95% CI: 9:13] in Israel, and 10% [95% CI: 8:14] in Canada.

## Publication bias

In this study, we assessed the publication bias by visually inspection of the funnel plot and conducting the Egger's test, which helps detect small-study effects (Fig 7). The funnel plot exhibits

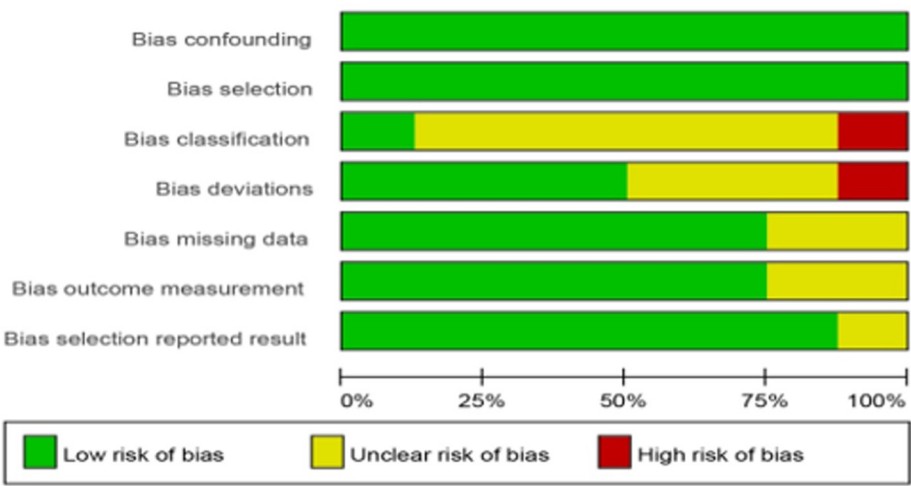

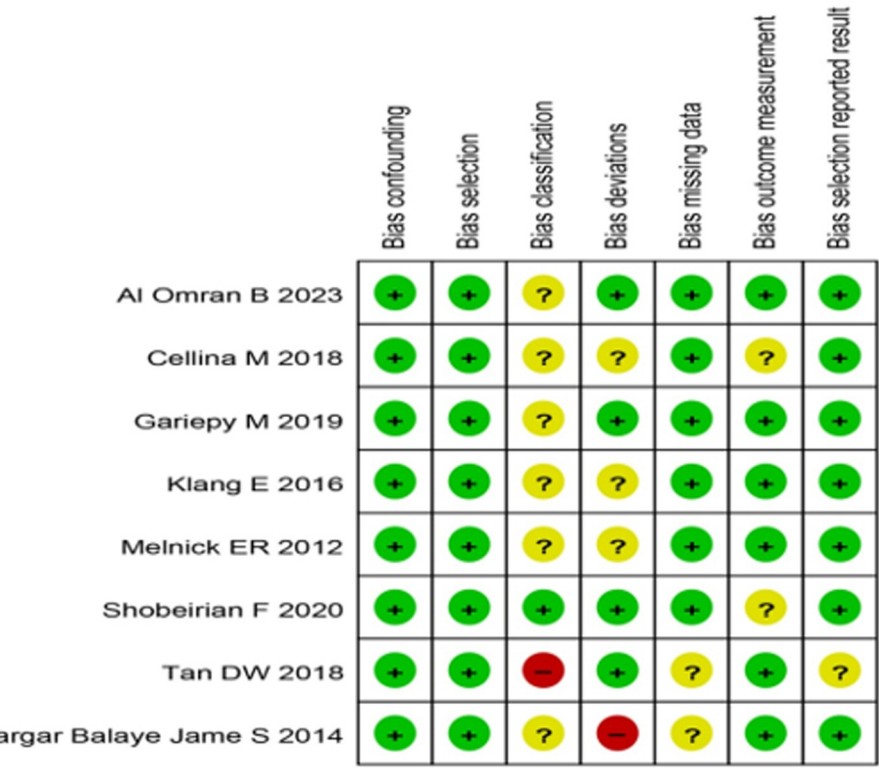

**Fig 2. Summary of risk of bias.** We assessed risk of bias in included studies utilizing the "Risk Of Bias In Non-randomised Studies - of Interventions" (ROBINS-I) tool. In studies, low risk of bias was observed in confounding and participant selection. One study had low bias in intervention classification, while others had uncertain or high risk. Bias in deviations from intended interventions varied, and most studies had low risk. Bias related to missing data and outcome measurement was generally low or unclear. Selection bias in reported results was low in most studies, with one unclear study.

asymmetry, and the presence of publication bias was not statistically significant: using Egger's test for small-study effects did not reach significance, as the bias coefficient was 16.76 [95% CI: [-7.09: 40.62] and the P-value 0.136. Also, we used Duval and Tweedie non-parametric trim and

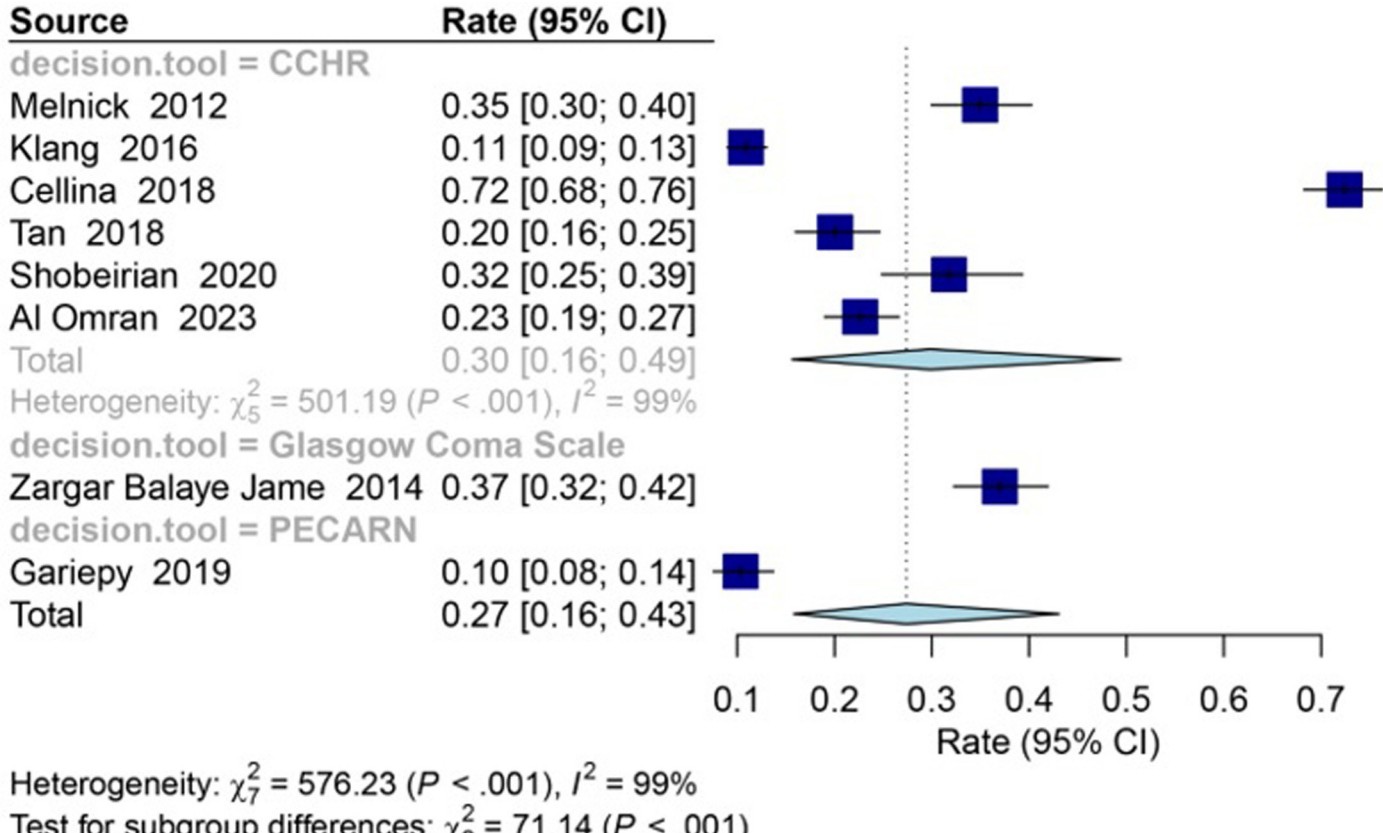

**Fig 3. Forest plot showing the pooled rate of overuse of computed tomography in mild head injury based on decision criteria.** Random effects model is used for analysis (95% confidence interval). The overall rate of CT overuse in mild head injury was estimated to be 27% [95% CI: 16–43; I$^2$ = 99%].

fill method. The adjusted rate from the trim and fill method did not show any significant difference compared to the unadjusted pooled rate estimates (rate = 27% [95% CI: 16–43; I$^2$ = 99%]).

## Discussion

This study aimed to investigate the rate of overuse of CT scans in patients with mild head injury. Our findings showed that the estimated rate of overuse was 27%. It is important to note that differences in study populations, including age, gender, comorbidities, and severity of head injury, may have influenced the rate of overuse observed in our study. Moreover, differences in study design, data collection, and analysis methods may be due to variations in healthcare systems, access to imaging technologies, insurance coverage, and physician practices [11]. Furthermore, our study found that the rate of overuse of CT scans varied between countries, which may reflect differences in healthcare policies and guidelines. This highlights the need for further investigation and the development of standardized guidelines to promote the appropriate use of CT scans in patients with mild head injury.

Our study findings revealed that the highest rate of CT scans ordered in the emergency department was based on the GCS criteria, accounting for 37% of cases. One possible explanation for this high rate is that some physicians may be overly cautious or risk-averse, leading to unnecessary CT scans being ordered [28]. Another contributing factor could be the lack of clear guidelines or standardized protocols for managing mild head injuries, resulting in wide variation in the use of CT scans among different healthcare providers [29]. This can lead to

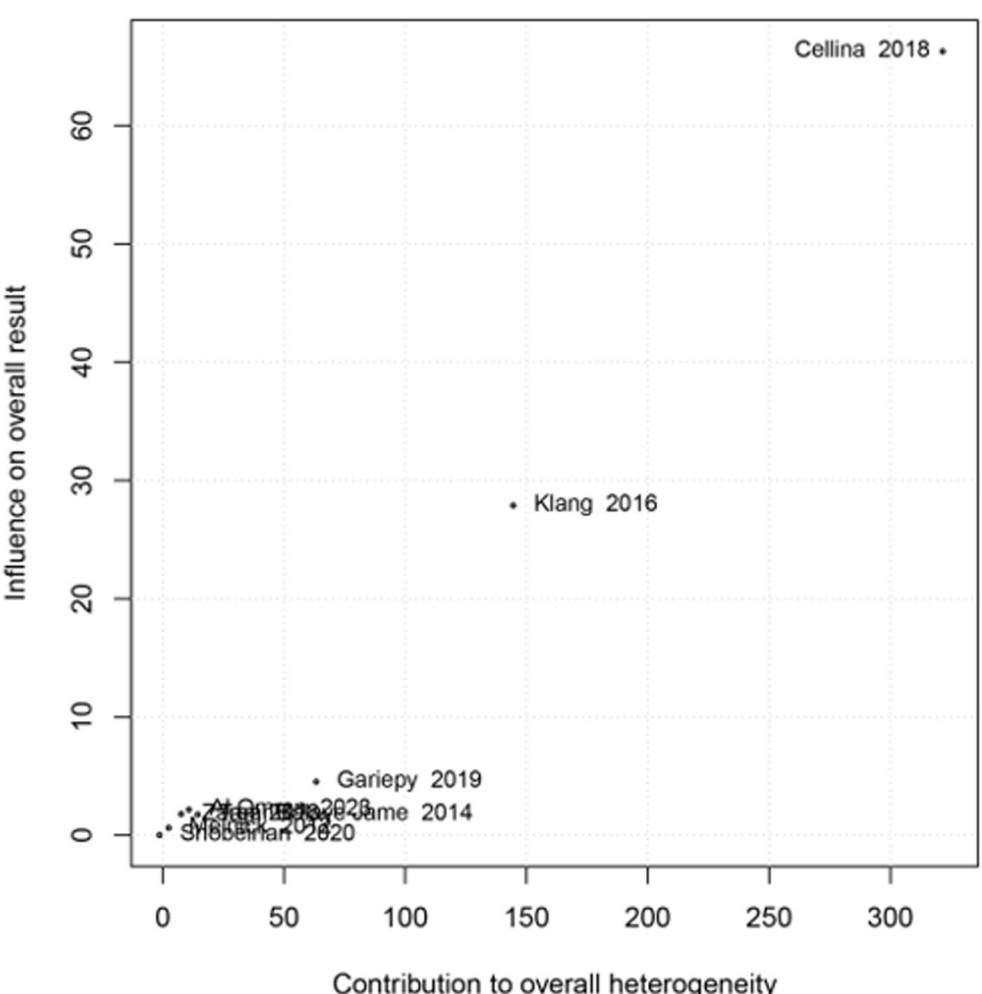

**Fig 4. Assessment of heterogeneity of included studies for overuse of computed tomography in mild head injury using.** Baujat plot. Most studies showed similar overuse rates, but one study (Cellina) had a significantly higher rate. Excluding it, the rate was 22% (95% CI: 15–32; I2 = 98%).

inconsistencies in clinical decision-making and may contribute to overuse [27]. It is crucial to address these issues and promote the appropriate use of CT scans to avoid unnecessary radiation exposure and reduce healthcare costs [30]. The development and implementation of evidence-based guidelines and protocols for managing mild head injuries could help standardize clinical practice and promote more appropriate use of CT scans [31]. Additionally, education and training for healthcare providers on the appropriate use of imaging and the risks of overuse may also be beneficial [32].

Our study also found that the highest rate of overuse according to clinical decision rules for CT scans was observed in the NICE guidelines, accounting for 43%. NICE guidelines are widely used in the UK, and their recommendations are considered evidence-based and authoritative [30]. Therefore, the high rate of overuse observed in these settings may indicate the need to reevaluate or revise the guidelines. One possible explanation for the high rate of overuse in NICE guidelines is that they may be overly cautious in recommending CT scans for patients with mild head injury [33]. The guidelines may not consider individual risk factors or clinical characteristics that may require further imaging, leading to a general recommendation

| Source | Rate (95% CI) |
|---|---|
| Omitting Melnick | 0.26 [0.14; 0.44] |
| Omitting Zargar Balaye Jame | 0.26 [0.14; 0.44] |
| Omitting Klang | 0.31 [0.18; 0.48] |
| Omitting Cellina | 0.22 [0.15; 0.32] |
| Omitting Tan | 0.29 [0.15; 0.47] |
| Omitting Gariepy | 0.31 [0.18; 0.47] |
| Omitting Shobeirian | 0.27 [0.14; 0.45] |
| Omitting Al Omran | 0.28 [0.15; 0.47] |
| Total | 0.27 [0.16; 0.43] |

**Fig 5. Forest plot of assessment the effect of excluding or including retained studies using sensitivity analysis.** In this analysis, each study is excluded and in its absence, the impact on the rate amount is evaluated. This analysis showed that the results did not change and the of CT overuse in mild head injury was estimated to be 27% [95% CI: 16–43].

for CT scanning for all patients with mild head injury [34]. Another potential factor contributing to differences in rate estimates is the use of different criteria to diagnose mild head injury in different studies [28,33,35]. This can result in differences in patient selection and ultimately the prevalence of overuse observed in each study. Furthermore, it is important to note that different countries may have different guidelines for the use of CT scans in mild head injury patients, leading to variations in practice patterns and rate estimates [36]. Therefore, it is crucial to develop evidence-based guidelines and protocols that consider individual risk factors

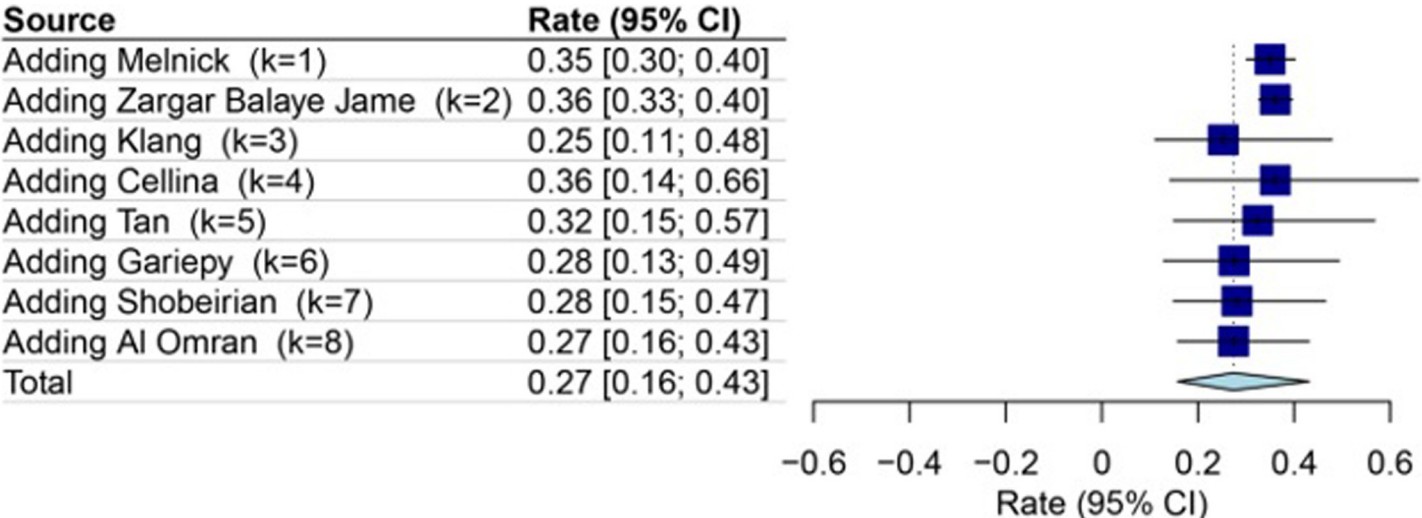

| Source | Rate (95% CI) |
|---|---|
| Adding Melnick (k=1) | 0.35 [0.30; 0.40] |
| Adding Zargar Balaye Jame (k=2) | 0.36 [0.33; 0.40] |
| Adding Klang (k=3) | 0.25 [0.11; 0.48] |
| Adding Cellina (k=4) | 0.36 [0.14; 0.66] |
| Adding Tan (k=5) | 0.32 [0.15; 0.57] |
| Adding Gariepy (k=6) | 0.28 [0.13; 0.49] |
| Adding Shobeirian (k=7) | 0.28 [0.15; 0.47] |
| Adding Al Omran (k=8) | 0.27 [0.16; 0.43] |
| Total | 0.27 [0.16; 0.43] |

**Fig 6. Forest plot to assess trends in rate of overuse of computed tomography in mild head injury changes over time using cumulative analysis.** The result of this analysis showed that the rate of CT overuse in mild head injury was 27% [95% CI: 16–43].

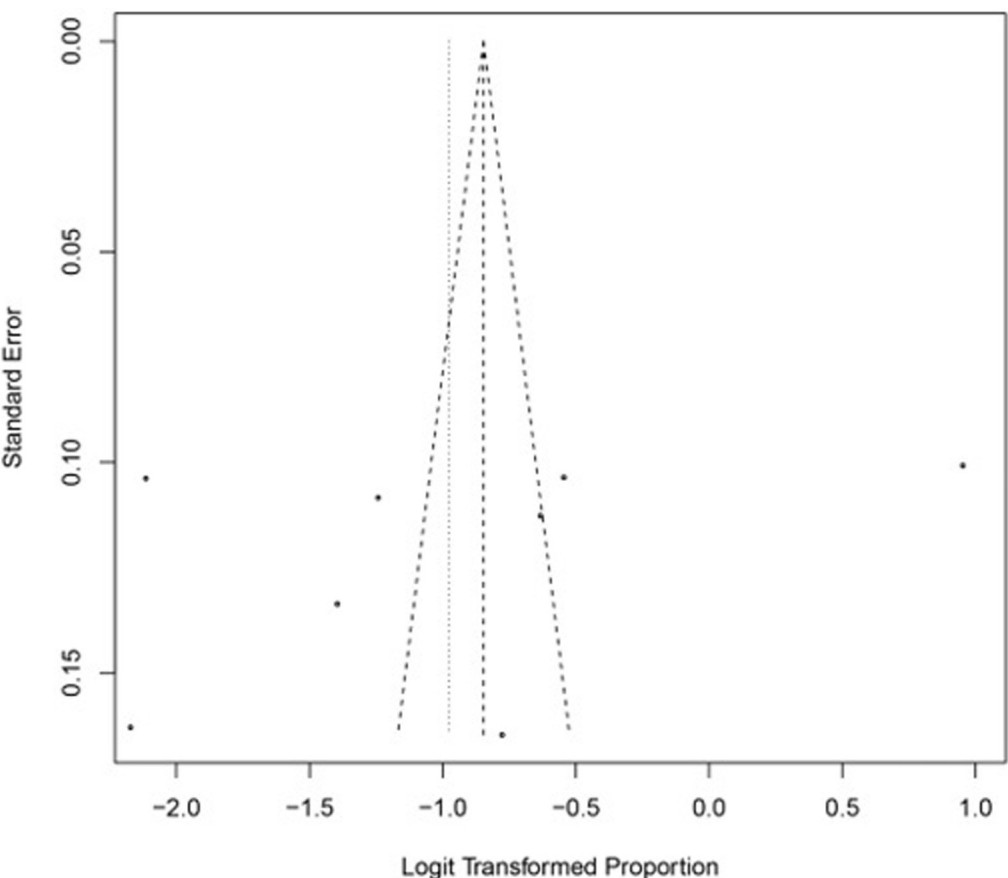

**Fig 7. Funnel plot for the assessment of publication bias among different studies.** The funnel plot exhibits asymmetry, and the presence of publication bias was not statistically significant: using Egger's test for small-study effects did not reach significance, as the bias coefficient was 16.76 [95% CI: [-7.09: 40.62] and the P-value 0.136.

and clinical characteristics to promote the appropriate use of CT scans in patients with mild head injury [25].

The rate of overuse was higher in CCHR than in NOC. One possible explanation for the higher rate of overuse in the CCHR is that the rule may be more sensitive but less specific in identifying patients who require CT scans [27]. The CCHR may include more criteria for ordering CT scans, leading to a higher rate of overuse, whereas the NOC may be more specific in identifying patients who require imaging, resulting in a lower rate of overuse which is consistent with the findings of the study of Stiell et al [28]. Another potential explanation for the difference in overuse between the CCHR and NOC could be due to differences in the populations in which the rules were developed and validated [37]. The CCHR was developed and validated in a Canadian population, whereas the NOC was developed and validated in a US population [24]. These populations may differ in terms of demographics, healthcare access, and other factors that may influence the rate of overuse of CT scans [26].

There exists a diverse array of decision rules recommended for triaging CT scans in this patient group, each one balancing varying sensitivities and specificities to detect serious injuries while minimizing the number of unnecessary CT scans. Moreover, certain decision rules have specific inclusion criteria, such as excluding anti-coagulated patients or being limited to those seeking medical attention within a specific timeframe after the injury. Additionally,

patients with concurrent cognitive impairment pose unique challenges in applying decision rules due to chronically impaired GCS scores, leading to differences in clinical decision-making regarding the necessity of imaging. It is essential to acknowledge the regional variation in the adoption of these decision rules, with the USA adopting a more liberal approach to CT imaging compared to the UK, where NICE guidelines are recommended. However, it should be emphasized that the contextual nature of overuse warrants attention, as even in the absence of guideline indications for imaging, certain situations may still warrant appropriate use of CT scans. In settings with limited CT imaging availability, triaging based on GCS alone might be considered, although this approach may not be clinically appropriate in resource-rich settings based in North America and Europe. Thus, our discussion highlights the significance of understanding the nuances surrounding overuse and inappropriate CT imaging in mild head injuries, emphasizing the importance of context-specific decision-making and the need for further research to establish more tailored guidelines for different healthcare settings.

The study investigated the rate of CT overuse in mild head injury patients based on age groups. The results revealed that individuals aged 40 years and below had a higher rate of CT overuse (36%) compared to those above 40 years (20%). Older patients may present with different symptoms or comorbidities that make clinicians less likely to order a CT scan for mild head injury [38]. Older patients may have a history of previous head injury, which may influence the decision-making process for ordering a CT scan [36]. Additionally, older patients may have more chronic health conditions or take medications that increase the risk of bleeding or other complications from a CT scan, leading clinicians to be more selective in their use of imaging. It is possible that older patients may have a lower threshold for accepting or declining a CT scan compared to younger patients [28]. Older patients may be more likely to have concerns about radiation exposure or other risks associated with CT scans, and may therefore be more hesitant to undergo unnecessary imaging [39]. Another possible explanation could be related to differences in the decision-making process among healthcare providers. Younger patients may be evaluated by healthcare providers who are more comfortable with clinical assessments, such as physical exams and history-taking, and may be less likely to order CT scans unless absolutely necessary [40]. In contrast, healthcare providers who evaluate older patients may be more cautious and tend to order more imaging studies due to concerns about potential complications.

Patient or family pressure, fear of litigation, and limited access to alternative diagnostic methods, such as MRI or ultrasound, are recognized factors that can contribute to the overuse of CT scans [6]. Furthermore, some emergency departments may lack the necessary resources, such as trained personnel or suitable equipment, to perform alternative diagnostic tests, leading to an over-reliance on CT scans as the primary diagnostic tool [35]. These factors may have contributed to the high rate of CT scan overuse observed in our study. To address the overuse of diagnostic interventions, healthcare providers must adopt evidence-based clinical guidelines and prioritize the appropriate use of diagnostic interventions based on each patient's individual circumstances [38]. This approach can help reduce unnecessary testing, limit the potential for harm, and improve patient outcomes. Healthcare providers must also engage in shared decision-making with patients to ensure that the risks and benefits of any diagnostic intervention are fully discussed and understood [30]. By doing so, patients can make informed decisions about their healthcare, and healthcare providers can ensure that they are providing high-quality care that is tailored to each patient's individual needs [25]. Healthcare systems must work to incentivize appropriate use of diagnostic interventions by promoting value-based care and implementing policies that discourage the overuse of diagnostic interventions.

There are several policies and strategies that can be implemented to reduce the overuse of CT scans in patients with mild head injuries:

Use validated clinical decision rules: Clinical decision rules such as the CCHR and NOC can help clinicians identify patients who are at low risk of intracranial injury and can safely forego CT scanning.

Attend training courses: Educating clinicians and patients about the risks and benefits of CT scans, and the potential harms of unnecessary radiation exposure, can help reduce overuse. This can include promoting the use of alternative imaging modalities or observation periods for patients who are at low risk of intracranial injury.

Develop local guidelines and protocols: Developing local guidelines and protocols can help standardize practice and reduce variations in the use of CT scans. This can include establishing clear criteria for CT scans, such as indications for imaging, the timing of scans, and follow-up plans.

Implement decision support tools: Decision support tools such as electronic medical record alerts and computerized clinical decision support systems can help remind clinicians about appropriate imaging indications and provide real-time feedback on imaging requests.

Implement audit and feedback mechanisms: Regularly auditing imaging requests and providing feedback to clinicians can help identify areas of overuse and opportunities for improvement.

Increase access to alternative imaging modalities: Increasing access to alternative imaging modalities such as MRI or ultrasound can provide clinicians with additional tools to evaluate patients with mild head injuries and reduce the reliance on CT scans.

Encourage shared decision-making: Shared decision-making between clinicians and patients can help ensure that patients are fully informed about the risks and benefits of CT scans, and that their preferences and values are taken into account when making imaging decisions.

Develop quality improvement initiatives: Quality improvement initiatives, such as team-based approaches to care and multidisciplinary care teams, can help reduce overuse of CT scans by improving communication and collaboration between healthcare providers.

Foster a culture of appropriate, responsible imaging: Establishing a culture of appropriate imaging within healthcare organizations can help promote responsible use of imaging technologies, make health systems more sustainable, and encourage clinicians to prioritize patient-centered care over unnecessary testing.

## Strengths and limitations

The strength of our meta-analysis lies in its global investigation of CT scan overuse specifically in patients with mild acute head injuries, a topic that has not been previously explored in the literature. To the best of our knowledge, this is the first study to evaluate the overuse of CT scans in mild head injuries in the world. Also, the comprehensive search of different databases was one of the strengths of this study.

There are several possible limitations of this manuscript that should be considered. Firstly, the limited number of studies included in the final analysis may affect the generalizability of the findings. Secondly, another shortcoming is given by the high heterogeneity observed in the included studies. To investigate possible sources of heterogeneity across studies, we performed

subgroup analysis based on subjects' age, patient assessment criteria, and study geographic region. Additionally, the lack of data on the possible causes of the high rate of overuse limits our ability to draw conclusions regarding the underlying factors contributing to this issue. Finally, it is important to note that this study focused specifically on the rate of overuse of CT scans for mild head injury and did not evaluate the potential harm or benefits associated with this practice.

## Conclusion

The rate of overuse of CT scans in mild head injury patients is rather high, with a computed figure of 27%. This suggests that there is a need for greater awareness and education among healthcare providers about appropriate imaging strategies Furthermore, efforts should be made to implement evidence-based guidelines that recommend the use of CT scans only in select cases, to reduce unnecessary exposure to radiation and optimize healthcare resource utilization.

### Protocol and registration

The study protocol of this review was registered in PROSPERO under the identification code CRD42023416080 [41].

### Supporting information

**S1 Fig. The overuse of computed tomography for mild head injury according to the region.** Random effects model used for analysis (95% confidence interval). The rate of CT overuse in mild head injury patients was 36% [95% CI: 3–92; $I^2$ = 100%] in Europe, 27% [95% CI: 20–36; $I^2$ = 91%] in Asia, and 20% [95% CI: 5–53; $I^2$ = 98%] in America.
(DOCX)

**S2 Fig. The overuse of computed tomography for mild head injury according to age.** Random effects model used for analysis (95% confidence interval). The rate of CT overuse in mild head injury patients was 36% [95% CI: 14–66; $I^2$ = 99%] in individuals aged 40 years and below, and 20% [95% CI: 13–30; $I^2$ = 92%] in those above 40 years.
(DOCX)

**S3 Fig. The overuse of computed tomography for mild head injury by the Canadian *computed tomography* Head Rule (*CCHR*).** Random effects model used for analysis (95% confidence interval). The overall rate of CT overuse in mild head injury according to the CCHR was estimated to be 30% [95% CI: 16–49; $I^2$ = 99%].
(DOCX)

**S4 Fig. The overuse of computed tomography for mild head injury by the National Institute for Health and Clinical Excellence (NICE).** Random effects model used for analysis (95% confidence interval). The overall rate of CT overuse in mild head injury according to the NICE was estimated to be 43% [95% CI: 13–80; $I^2$ = 99%].
(DOCX)

**S5 Fig. The overuse of computed tomography for mild head injury by the New Orleans Criterion (NOC).** Random effects model used for analysis (95% confidence interval). The overall rate of CT overuse in mild head injury according to the NOC was estimated to be 18% [95% CI: 5–18; $I^2$ = 97%].
(DOCX)

**S6 Fig. The overuse of computed tomography for mild head injury based on country.** Random effects model used for analysis (95% confidence interval). The rate of CT Overuse of scan in patients with mild head injury was 35% [95% CI: 30:40] in America, 35% [95% CI: 30:40] in Iran, 23% [95% CI: 19:27] in Bahrain, 20% [95% CI: 16:25] in Singapore, 11% [95% CI: 9:13] in Israel and 10% [95% CI: 8:14] in Canada.
(DOCX)

**S1 Table. Preferred Reporting Items for Systematic Reviews and Meta-Analyses (PRISMA) checklist.**
(DOCX)

**S2 Table. PRISMA 2020 for abstracts checklist.**
(DOCX)

**S3 Table. The search strategy.**
(DOCX)

**S4 Table. Quality assessment of the included studies using the Newcastle-Ottawa Scale (NOS).**
(DOCX)

## Author Contributions

**Conceptualization:** Maryam Saran, Mehrdad Gholami, Samad Azari.

**Data curation:** Morteza Arab-Zozani, Mehrdad Gholami, Samad Azari, Masoud Behzadifar.

**Formal analysis:** Samad Azari, Masoud Behzadifar.

**Investigation:** Meysam Behzadifar.

**Methodology:** Morteza Arab-Zozani, Masoud Behzadifar.

**Project administration:** Masoud Behzadifar.

**Resources:** Nicola Luigi Bragazzi, Masoud Behzadifar.

**Supervision:** Maryam Saran, Masoud Behzadifar.

**Validation:** Meysam Behzadifar, Samad Azari, Masoud Behzadifar.

**Writing – original draft:** Maryam Saran, Morteza Arab-Zozani, Nicola Luigi Bragazzi, Masoud Behzadifar.

**Writing – review & editing:** Maryam Saran, Nicola Luigi Bragazzi, Masoud Behzadifar.

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
