## [Decision Letter · Decision Letter 0]

12 Jul 2023

PONE-D-23-17711Overuse of computed tomography for mild head injury: a systematic review and meta-analysisPLOS ONE

Dear Dr. Behzadifar,

Thank you for submitting your manuscript to PLOS ONE. After careful consideration, we feel that it has merit but does not fully meet PLOS ONE’s publication criteria as it currently stands. Therefore, we invite you to submit a revised version of the manuscript that addresses the points raised during the review process.

We look forward to receiving your revised manuscript.

Kind regards,

Jafar Kolahi

Academic Editor

PLOS ONE

Additional Editor Comments:

Please take note of the following corrections:

1. Perform a risk of bias assessment and present the results in a 'bias graph'.

2. Conduct a publication bias analysis and present the results in a 'funnel plot'.

3. Apply the trim-and-fill method to estimate potentially missing studies due to publication bias in the funnel plot and adjust the overall effect estimate accordingly.

Reviewers' comments:

Reviewer's Responses to Questions

**Comments to the Author**

1. Is the manuscript technically sound, and do the data support the conclusions?

Reviewer #1: Yes

Reviewer #2: Yes

Reviewer #3: Partly

2. Has the statistical analysis been performed appropriately and rigorously? 

Reviewer #1: I Don't Know

Reviewer #2: Yes

Reviewer #3: No

3. Have the authors made all data underlying the findings in their manuscript fully available?

Reviewer #1: Yes

Reviewer #2: No

Reviewer #3: Yes

4. Is the manuscript presented in an intelligible fashion and written in standard English?

Reviewer #1: Yes

Reviewer #2: Yes

Reviewer #3: Yes

5. Review Comments to the Author

Reviewer #1: Dear authors,

The present study is a systematic review and meta-analysis on the overuse of CT scanning in mild head trauma. I appreciate the authors’ efforts for conducting this review on such an interesting and important topic. I find the manuscript generally well-written. But based on my review, there are some issues that need to be revised and addressed:

- There are several grammatical errors and typos throughout your manuscript. The manuscript may benefit from English editing by a native user.

- Quality assessment and risk of bias assessment are two separate concepts.

- Some abbreviations in the text are not necessary: LMIC

- Please elaborate further on the justification of your study.

- A brief literature review is missing.

- Inclusion and exclusion criteria need revision.

Reviewer #2: Thank you for the opportunity to review this well conducted and thorough systematic review which addresses an important clinical issue—use of CT imaging in mild traumatic brain injury.

My main concern is how overuse or inappropriate CT imaging in this group can be defined. There are a range of decision rules which can be/are recommended to be used to triage CT imaging in this group. They all balance differing sensitivities and specificities (i.e. the risk of missing a serious injury against the number of negative CT scans need to identify a rare injury) and have different inclusion criteria (e.g. the CCHR excludes anti-coagulated patients and NICE guidelines is not applicable to patient attending > 24 hours after injury). There are also patients with concurrent cognitive impairment where application of a decision rule is difficult due to chronically impaired GCS and clinical decision-making re need for imaging may differ.

Based on this different decision rules are recommended for use in different settings and the USA has a more liberal approach to CT imaging than the UK where the NICE guidelines are recommended. In North America and Europe triage of CT imaging based on GCS alone would not be clinically appropriate but this may be acceptable in other setting with more limited availability of CT imaging.

I think this nuance needs to be brought out in the discussion as over use is contextual and even where there is no guideline indication for imaging it still may be appropriate.

Reviewer #3: The manuscript entitle “Overuse of computed tomography for mild head injury: a systematic review and meta-analysis” was interesting and showed a high rate of using CT for a mild head injury, hence it might be useful for policymakers (developing guideline). However; many revisions should be done, particularly in the method and the result parts. Please see the attached file.

6. PLOS authors have the option to publish the peer review history of their article (what does this mean?). If published, this will include your full peer review and any attached files.

Reviewer #1: No

Reviewer #2: **Yes: **Carl Marincowitz

Reviewer #3: No

---

## [Author Response · Author response to Decision Letter 0]

14 Aug 2023

Dear editor

I wanted to take a moment to express my sincere gratitude for your valuable comments and feedback on my manuscript for the journal. Your thoughtful and constructive input has helped to significantly improve the quality and clarity of the manuscript. I appreciate the time and effort that you put into reviewing the manuscript and providing such insightful comments. As you may be aware, we have carefully considered your feedback and have provided responses to each of the comments raised by the reviewers. Your expertise and attention to detail have been invaluable in helping to shape the final version of the manuscript. We hope that the revised manuscript meets the required standards and addresses all of your concerns.

Thank you once again for your dedication to the peer review process and for your contributions to the advancement of scientific knowledge.

Sincerely,

Dr. Masoud Behzadifar

Additional Editor Comments:

Please take note of the following corrections:

1. Perform a risk of bias assessment and present the results in a 'bias graph'. Thank you so much for your comment. We resvised and done.

2. Conduct a publication bias analysis and present the results in a 'funnel plot'. Thank you so much for your comment. We resvised and done. 3. Apply the trim-and-fill method to estimate potentially missing studies due to publication bias in the funnel plot and adjust the overall effect estimate accordingly. Thank you so much for your comment. We resvised and done.

Reviewer #1: Dear authors,

The present study is a systematic review and meta-analysis on the overuse of CT scanning in mild head trauma. I appreciate the authors’ efforts for conducting this review on such an interesting and important topic. I find the manuscript generally well-written. But based on my review, there are some issues that need to be revised and addressed:

- There are several grammatical errors and typos throughout your manuscript. The manuscript may benefit from English editing by a native user. Thank you so much for your comment. We resvised and done.

 - Quality assessment and risk of bias assessment are two separate concepts. Thank you so much for your comment. We resvised and done.

- Some abbreviations in the text are not necessary: LMIC. Thank you so much for your comment. We resvised and done.

- Please elaborate further on the justification of your study. Thank you so much for your comment. We resvised and done.

- Inclusion and exclusion criteria need revision. Thank you so much for your comment. We resvised and done.

Reviewer #2: Thank you for the opportunity to review this well conducted and thorough systematic review which addresses an important clinical issue—use of CT imaging in mild traumatic brain injury.

My main concern is how overuse or inappropriate CT imaging in this group can be defined. There are a range of decision rules which can be/are recommended to be used to triage CT imaging in this group. They all balance differing sensitivities and specificities (i.e. the risk of missing a serious injury against the number of negative CT scans need to identify a rare injury) and have different inclusion criteria (e.g. the CCHR excludes anti-coagulated patients and NICE guidelines is not applicable to patient attending > 24 hours after injury). There are also patients with concurrent cognitive impairment where application of a decision rule is difficult due to chronically impaired GCS and clinical decision-making re need for imaging may differ.

Based on this different decision rules are recommended for use in different settings and the USA has a more liberal approach to CT imaging than the UK where the NICE guidelines are recommended. In North America and Europe triage of CT imaging based on GCS alone would not be clinically appropriate but this may be acceptable in other setting with more limited availability of CT imaging.

I think this nuance needs to be brought out in the discussion as over use is contextual and even where there is no guideline indication for imaging it still may be appropriate. Thank you so much for your comment. We resvised and done.

Reviewer #3: The manuscript entitle “Overuse of computed tomography for mild head injury: a systematic review and meta-analysis” was interesting and showed a high rate of using CT for a mild head injury, hence it might be useful for policymakers (developing guideline). However; many revisions should be done, particularly in the method and the result parts. Please see the attached file.

The manuscript entitle “Overuse of computed tomography for mild head injury: a systematic review and meta-analysis” was interesting and showed a high rate of using CT for a mild head injury, hence it might be useful for policymakers (developing guideline). However; many revisions should be done, particularly in the method and the result parts. 

Title: -

Abstract

1. Write it according to PRISMA 2020 for Abstracts Checklist (many revision are needed). Thank you so much for your comment. We write abstract according PRISMA 2020. 

2. Apply the other comments of text, if applicable. Thank you so much for your comment. We resvised and done.

Keywords

1. Write it based on MeSH. Thank you so much for your comment. We resvised keywords based MeSH. 

Introduction

1. This part was lengthy and needs some revision. Thank you so much for your comment. We shorten it. 

Introduction, 2nd paragraph

1. The authors wrote about the management and consequence of the trauma. While the main aim of the study was the diagnosis. Delete it. Thank you so much for your comment. We deleted. 

Introduction, 3rd and 4th paragraphs

1. Mix these paragraphs and summarize them. Thank you so much for your comment. We mixed paragraphs. 

Introduction, 5th paragraphs

“as reported in studies conducted across different countries” in the aim part: I think this issue was not the main aim of your studies. Indeed, there were no subgroups regarding the region. Thank you so much for your comment. We analyzed based region and added in the text. 

Method

This part needs many revisions and responses. See the comments, please. 

1. Certainty (quality) of evidence should conduct. Thank you so much for your comment. We revised and done. 

2. A table of findings should be added. Thank you so much for your comment. We added. 

Protocol and Registration

1. Any deviation should mention here. For example: review question#2: “What is the pooled prevalence of computed tomography overuse for mild head injury based on region?” I did not find any subgroup regarding the region. Thank you so much for your comment. We analyzed based region and added in the text. 

Study Design: 

1. I could not find the design of study in this paragraph. Thank you so much for your comment. We deleted this phrase. 

2. For PRISMA 2020 Checklist, you should report the page in the checklist in the supplementary file. Thank you so much for your comment. We added number pages. 

3. Add the PRISMA 2020 for Abstracts Checklist in the supplementary file too. Thank you so much for your comment. We added PRISMA for abstract in the supplementary. 

Search Strategy and Selection Criteria

1. Why did the author set a time limitation (January 2000)? Thank you so much for your comment. We revised in text (from inception until April 1, 2023).

2. Add the page of the appendix. Thank you so much for your comment. We added number pages

3. “two team”. Add the initial name of the researchers. Thank you so much for your comment. We added. 

4. “to find additional studies”. Delete it. Thank you so much for your comment. We deleted. 

5. "studies were published in peer-reviewed journals" was one of the inclusion criteria. Please explain that, while you should have considered the published article, you searched gray literature. More details should be added around gray literature. Thank you so much for your comment. We revised. 

Eligibility Criteria

1. When some data were missed/ full texts were unavailable, it should be supposed to contact the corresponding author. Please explain why the author did not contact the corresponding authors. Thank you so much for your comment. We did this case and added in text. 

2. “those published as reviews”. Delete it. Previously it was mentioned that the observational study designs were included. Thank you so much for your comment. We deleted. 

3. It is unclear whether the authors consider or not considered unpublished records. Thank you so much for your comment. We added.

4. “intervention studies, or studies in languages other than English.” The type and language limitations were mentioned, previously. Thank you so much for your comment. We deleted. 

Data Collection

1. “two author teams extracted information”. Add the initial name of the researchers. Thank you so much for your comment. We added.

2. Gender ratio or number? Thank you so much for your comment. We added number.

3. “criteria for CT scan performance, and prevalence of overuse”. I could not find these variables in the table. Thank you so much for your comment. We dded in Table 1. 

4. “resolved by a senior researcher”. Add the initial name of the researchers. Thank you so much for your comment. We added.

5. “a third reviewer was introduced”. Add the initial name of the researcher. Thank you so much for your comment. We added.

6. “designed by the authors' group”. All authors or some of them (if some, add the initial name of them). Thank you so much for your comment. We added.

Risk of Bias of Individual Studies

7. How many authors did that? Add the initial name of them. Thank you so much for your comment. We added.

Statistical Analysis

8. "inverse variance" and "Dersimonian-Laird" are two different models. Why did the authors use two different computations? Thank you so much for your comment. We cahnged.

1. Why you did not conduct a sensitivity analysis by omitting high RoB studies? What was the benefit of conducting a RoB assessment? Thank you so much for your comment. We cahnged this section.

2. mean of age. Thank you so much for your comment.We changed.

3. I recommend conducting quantitative publication measurement (Regression-based Egger/Harbord/Peters test). In addition, funnel plots could be add in supplementary file. Thank you so much for your comment. We added and changed. 

Results

1. Many revisions are needed.

Study Selection

1. “and 45 were excluded.” According to PRISMA checklist 2020, 16b: Cite studies that might appear to meet the inclusion criteria, but which were excluded, and explain why they were excluded. Hence, cite the articles and explain the reason for the exclusion in a table in the supplementary file. Thank you so much for your comment. We added in the supplementary file. 

2. “Figure 1”. Use PRISMA 2020 flow diagram for new systematic. Thank you so much for your comment. We changed based PRISMA 2020 flow diagram.

Study Characteristics

9. “Out of a total of 8 articles, two were done in Iran. The rest of the countries had assigned one article each, and”. Delete it. Thank you so much for your comment. We deleted.

10. Add more details regard the “criteria for CT scan performance” and “criteria for diagnosing acute head injury”. Thank you so much for your comment. All patients had mild acute head injury.

Risk of Bias

11. The visualization cloud makes this part easier for the reader. Thank you so much for your comment. We added figures in the text. 

12. More details are needed: for example, which domain was high risk among the studies? Add results for each study and each domain. Thank you so much for your comment. We added.

13. “The risk of bias of the included studies was assessed through the use of the NOS checklist.” Delete it. Thank you so much for your comment. We deleted..

1. “One study received a score of 6, two studies scored 7, four studies scored 8,” Cite the studies. Thank you so much for your comment. We added cite.

Synthesis of Results

1. “using a random model”. Delete it. Thank you so much for your comment. We deleted.

2. Write the I2 in the bracket and delete “This was observed with I2=99%”. Hence, the correct from is [95% CI: 16-43; I2=99%]. Thank you so much for your comment. We deleted and corrected. 

3. “Figure 2”. Delete this figure. Figure 3 was enough. Thank you so much for your comment. We changed.

4. “To ensure the robustness of the sensitivity analysis, the impact of each study was consistent before and after the analysis. Furthermore, the effect of study order was examined through cumulative meta-analysis.” Here, you should report the result here. Thank you so much for your comment. We done this comment. 

5. “Here, you should report the result here”. In the method part, any subgroup should be mentioned. Thank you so much for your comment. We done this comment. 

6. Add I2 when reporting 95% CI. Thank you so much for your comment. We done this comment. 

7. “Accordance clinical decision rules”. In the method part, any subgroup should be mentioned. These variables are also not reported in data extraction table 1. Thank you so much for your comment. We done this comment. 

Comparison of criteria used in included studies

8. This part was not clear. In addition, neither declare in the aim nor method part. I think it should be deleted. Thank you so much for your comment. We deleted. 

Meta-regression (last paragraph of result)

1. “This part was not clear. In addition, neither declare in the aim nor method part. I think it should be deleted.” Delete it. Thank you so much for your comment. We deleted.

2. “The results are presented in Table 2”. The results of meta-regression are presented in Table Thank you so much for your comment. We deleted.

Discussion

1. Some parts of the discussion should reassess after the method and result revisions.

Discussion, 1st paragraph 

1. “However, the rate range reported in the 8 studies included in the final analysis varied.” Delete it. Thank you so much for your comment. We deleted.

2. “It is important to note that differences in study populations, including age, gender, comorbidities, and severity of head injury, may have influenced the rate of overuse observed in our study”. Some of these variables were available in your studies and could be checked (or checked) through mete-regression to support your discussion. Thank you so much for your comment. We deleyed the meta-regression based reviewer comment. 

3. “Furthermore, our study found that the rate of overuse of CT scans varied between countries,” you could do a subgroup based on country to support your discussion. Thank you so much for your comment. We analyzed base country and added in the results.

Discussion, 2nd paragraph 

1. “Glasgow Coma Scale (GCS)”. Delete “Glasgow Coma Scale”. Previously you define this abbreviation. Thank you so much for your comment. We deleted. 

Discussion, 3rd paragraph: - 

Discussion, 4th paragraph

1. It needs to reassess after revision or the author's response. 

Discussion, 5th paragraph

1. First line, “our study found that”. Delete it. Thank you so much for your comment. We deleted. 

Discussion, 6th paragraph: -

Discussion, 7th paragraph

1. This part (policies were suggested) was a good point of this study.

Strengths and Limitations

1. Add more strength regarding the work. Thank you so much for your comment. We added and changed.

2. How did the authors deal with heterogeneity? Thank you so much for your comment. We used subgroup analysis. 

Conclusion

1. “Based on the findings of our meta-analysis, it can be concluded that”. Delete it. Thank you so much for your comment. We deleted. 

2. “for mild head injury patients”. Delete it. Thank you so much for your comment. We deleted.

3. “Future research should aim to identify barriers and facilitators to the implementation of such guidelines, and to evaluate the impact of interventions aimed at reducing excessive CT scan use in mild head injury patients.” Delete it. Thank you so much for your comment. We deleted.

Data Availability

1. Data relevant to the meta-analysis were not in the supplementary file. Thank you so much for your comment. We added file. 

Abbreviations

1. “CI: confidence intervals”. Write it as “CI: Confidence intervals”. Check all words. Thank you so much for your comment. We checked and revised. 

Funding

1. “The author(s) received no financial support for the research, authorship, and/or publication of this article.” In CRD, Lorestan University of Medical Sciences was mentioned for funding. Transparency is very important for each study. Thank you so much for your comment. We changed this in the CRD. 

Author Contributions

2. Some authors were added to the paper while their names were not in the CRD protocol. I recommend, if any deviation from protocol was done, revise the protocol in CRD. Thank you so much for your comment. We changed this in the CRD. 

3. While the authors declare "no funding", what were they mean by mentioning resources?. Thank you so much for your comment. We changed this in the CRD. 

1. 

References:-

Table 1

1. In “First author” column: Delete the initial names. Thank you so much for your comment. We deleted. 

2. “Glasgow coma scale”. Write it “Glasgow Coma Scale”. Thank you so much for your comment. We revised. 

3. “Age”. Add the value of this variable. Thank you so much for your comment. We revised. 

4. “NA”. Define the NA in the footnote. Thank you so much for your comment. We revised. 

5. “the Pediatric Emergency Care Applied Research Network (PECARN)” in footnote: write as “PECARN: the Pediatric Emergency Care Applied Research Network”. Thank you so much for your comment. We revised. 

Table 2: -

Figure 1

6. Use PRISMA 2020 flow diagram. Thank you so much for your comment. We revised. 

7. What was the difference between the two last lines in excluded box? Thank you so much for your comment. We revised. 

Figure 2

8. Delete this figure. Figure 3 was enough. Thank you so much for your comment. We deleted. 

Figure 3. 

1. Delete the initial names. Thank you so much for your comment. We deleted.

Figure 4, 5 and 6

1. As previously mentioned, this part (Comparison of criteria used in included studies) was not clear. Thank you so much for your comment. We deleted.

Supporting Information

Search strategy 

1. Add “PRESS Guideline — Search Submission & Peer Review Assessment”

PRISMA Checklist 

9. For PRISMA 2020 Checklist, you should report the page in the checklist in the supplementary file. Thank you so much for your comment. We revised. 

10. Add the PRISMA 2020 for Abstracts Checklist in the supplementary file too. Thank you so much for your comment. We revised.

---

## [Editor Report · Decision Letter 1]

30 Aug 2023

PONE-D-23-17711R1Overuse of computed tomography for mild head injury: a systematic review and meta-analysisPLOS ONE

Dear Dr. Behzadifar,

Thank you for submitting your manuscript to PLOS ONE. After careful consideration, we feel that it has merit but does not fully meet PLOS ONE’s publication criteria as it currently stands. Therefore, we invite you to submit a revised version of the manuscript that addresses the points raised during the review process.

We look forward to receiving your revised manuscript.

Kind regards,

Jafar Kolahi

Academic Editor

PLOS ONE

Journal Requirements:

Additional Editor Comments:

Please add sensitivity test, Baujat test and related plots.

---

## [Author Response · Author response to Decision Letter 1]

30 Aug 2023

Dear editor

I wanted to take a moment to express my sincere gratitude for your valuable comments and feedback on my manuscript for the journal. Your thoughtful and constructive input has helped to significantly improve the quality and clarity of the manuscript. I appreciate the time and effort that you put into reviewing the manuscript and providing such insightful comments. As you may be aware, we have carefully considered your feedback and have provided responses to each of the comments raised by the reviewers. Your expertise and attention to detail have been invaluable in helping to shape the final version of the manuscript. We hope that the revised manuscript meets the required standards and addresses all of your concerns.

Thank you once again for your dedication to the peer review process and for your contributions to the advancement of scientific knowledge.

Sincerely,

Dr. Masoud Behzadifar

Additional Editor Comments:

Please take note of the following corrections:

Please add sensitivity test, Baujat test and related plots. 

Thank you so much for your comment. We added Baujat test and sensitivity test and related plots (Figure 4 and Figure 5).

---

## [Editor Report · Decision Letter 2]

20 Sep 2023

PONE-D-23-17711R2Overuse of computed tomography for mild head injury: a systematic review and meta-analysisPLOS ONE

Dear Dr. Behzadifar,

Thank you for submitting your manuscript to PLOS ONE. After careful consideration, we feel that it has merit but does not fully meet PLOS ONE’s publication criteria as it currently stands. Therefore, we invite you to submit a revised version of the manuscript that addresses the points raised during the review process.

We look forward to receiving your revised manuscript.

Kind regards,

Jafar Kolahi

Academic Editor

PLOS ONE

Journal Requirements:

Additional Editor Comments:

Please add citations to the table of included articles.

---

## [Author Response · Author response to Decision Letter 2]

20 Sep 2023

Dear editor

I wanted to take a moment to express my sincere gratitude for your valuable comments and feedback on my manuscript for the journal. Your thoughtful and constructive input has helped to significantly improve the quality and clarity of the manuscript. I appreciate the time and effort that you put into reviewing the manuscript and providing such insightful comments. As you may be aware, we have carefully considered your feedback and have provided responses to each of the comments raised by the reviewers. Your expertise and attention to detail have been invaluable in helping to shape the final version of the manuscript. We hope that the revised manuscript meets the required standards and addresses all of your concerns.

Thank you once again for your dedication to the peer review process and for your contributions to the advancement of scientific knowledge.

Sincerely,

Dr. Masoud Behzadifar

Additional Editor Comments:

Please add citations to the table of included articles. Thank you so much for your comment. We added references to the table of included articles.

---

## [Editor Report · Decision Letter 3]

5 Oct 2023

PONE-D-23-17711R3Overuse of computed tomography for mild head injury: a systematic review and meta-analysisPLOS ONE

Dear Dr. Behzadifar,

Thank you for submitting your manuscript to PLOS ONE. After careful consideration, we feel that it has merit but does not fully meet PLOS ONE’s publication criteria as it currently stands. Therefore, we invite you to submit a revised version of the manuscript that addresses the points raised during the review process.

We look forward to receiving your revised manuscript.

Kind regards,

Jafar Kolahi

Academic Editor

PLOS ONE

Journal Requirements:

Additional Editor Comments:

Please extend figure legends.

---

## [Author Response · Author response to Decision Letter 3]

6 Oct 2023

Dear Dr. Jafar Kolahi

Academic Editor of the PLOS ONE

I would like to express my sincere gratitude for your time and valuable feedback on my manuscript submitted to PLOS ONE. I appreciate the thorough review of my work and the opportunity to revise it in accordance with your comments and suggestions. I have carefully considered of your comment and have made the necessary revisions to improve the quality and clarity of the manuscript. Below, I address each of your points and provide an explanation of the changes made. I believe that the revisions have significantly strengthened the manuscript and have addressed the concerns raised during the initial review. Additionally, I have taken care to ensure that the manuscript complies with the journal's guidelines and formatting requirements. Please find attached the revised manuscript, along with a marked-up version highlighting the changes made. I hope that you will find these revisions satisfactory, and they meet the standards of the PLOS ONE Journal.

With best wishes

Masoud Behzadifar

Correspondence author

Additional Editor Comments:

Please extend figure legends.

Thank you for your constructive feedback and your valuable suggestion to extend the figure legends in our manuscript. In response to your comment, we have revised the figure legends for all figures in the manuscript. We have expanded upon the information provided in the legends to provide more context and explanation of the key findings and elements depicted in each figure. These extended figure legends now provide a more comprehensive description of the data, methodology, and results associated with each figure.

---

## [Editor Report · Decision Letter 4]

16 Oct 2023

Overuse of computed tomography for mild head injury: a systematic review and meta-analysis

PONE-D-23-17711R4

Dear Dr. Behzadifar,

We’re pleased to inform you that your manuscript has been judged scientifically suitable for publication and will be formally accepted for publication once it meets all outstanding technical requirements.

Kind regards,

Jafar Kolahi

Academic Editor

PLOS ONE
---

## [Editor Report · Acceptance letter]

3 Jan 2024

PONE-D-23-17711R4 

PLOS ONE

Dear Dr. Behzadifar, 

I'm pleased to inform you that your manuscript has been deemed suitable for publication in PLOS ONE. Congratulations! Your manuscript is now being handed over to our production team.

Kind regards, 

on behalf of

Dr. Jafar Kolahi 

Academic Editor

PLOS ONE